# MEMORY-EFFICIENT FINE-TUNING VIA LOW-RANK ACTIVATION COMPRESSION

## ABSTRACT

The parameter-efficient fine-tuning paradigm has garnered significant attention with the advancement of foundation models. Although numerous methods have been proposed to reduce the number of trainable parameters, their substantial memory overhead remains a critical bottleneck that hinders practical deployment. In this paper, we observe that model activations constitute a major source of memory consumption, especially under large batch sizes and long context lengths; however, the rank of the activations remains consistently low. Motivated by this insight, we propose a memory-efficient fine-tuning approach *Low-Rank Activation Compression* (LORACT). Unlike prior work, LORACT provides a more flexible and versatile compressing strategy that can be applied online during the forward pass without the need for any calibration data. Moreover, LORACT incorporates a novel sampling-based orthogonal decomposition algorithm specifically designed for low-rank matrices, offering improved computational efficiency and a tighter error bound compared to the widely used RSVD. Experiments on both vision and language tasks demonstrate the effectiveness of LORACT. Notably, LORACT further reduces activation memory by approximately 80% in comparison with the widely adopted LoRA method, while maintaining competitive performance. The source code is available in the supplementary material.

## 1 INTRODUCTION

With the rapid development of foundation models, the parameter-efficient fine-tuning (PEFT) has emerged as a prevailing paradigm for adapting the models to downstream tasks (Houlsby et al., 2019; Lester et al., 2021; Li & Liang, 2021; Hu et al., 2022; Yang et al., 2024). The primary motivation of PEFT is to fine-tune foundation models using only a small set of trainable parameters. By this means, it aims to leverage minimal additional memory overhead while enhancing the adaptability to specific downstream tasks. However, despite reducing the number of trainable parameters, PEFT still incurs non-negligible memory consumption during fine-tuning, which significantly constrains its flexibility in practical scenarios (Simoulin et al., 2025; Huang et al., 2025).

In this paper, we systematically investigate the underlying reasons of the substantial memory consumption of the PEFT method. As illustrated in Figure 1, the memory costs associated with gradients and optimizer states are significantly reduced, which is owing to the reduction in the number of trainable parameters. However, the model activation memory constitutes the dominant factor of the total memory overhead, which scales approximately linearly with both batch size and context length. The substantial memory consumption is inconsistent with the original motivation of PEFT, which inevitably limits the practical deployment. Therefore, it is crucial to develop methods to advance parameter-efficient fine-tuning towards a memory-efficient process.

Prevailing memory-efficient fine-tuning methods primarily focus on the compression of gradients and optimizer states, such as Galore (Zhao et al., 2024), FLora (Hao et al., 2024), and Apollo (Zhu et al., 2024). However, these approaches are ill-suited for the PEFT setting, since PEFT already drastically reduces the number of trainable parameters, and the memory overhead from gradients and optimizer states is mostly manageable. Moreover, these methods neglect the problem of substantial activation memory. Despite some prior work has explored activation compression, these methods still suffer from multiple limitations. For instance, LoRA-FA (Zhang et al., 2023) and CompAct (Shamshoum et al., 2025) provide compression by confining the scope to specific layers, which

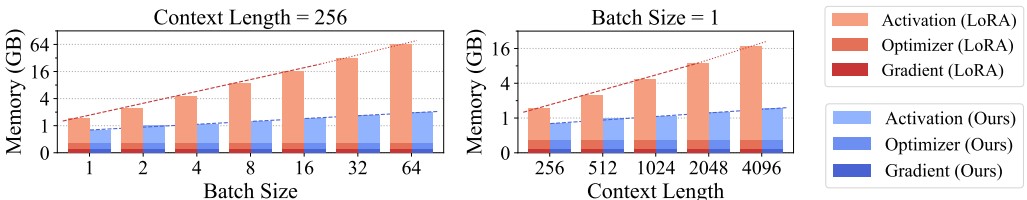

Figure 1: The additional memory cost for fine-tuning LLaMa2-7B with LoRA. The endpoints of the dashed line denote estimated results, as these configurations are infeasible on consumer-grade GPU.

may fail to address the full memory overhead. CoLA (Liu et al., 2025) and VeLoRA (Miles et al., 2024) employ sub-optimal compression algorithms that may significantly degrade performance. ESPACE (Sakr & Khailany, 2024) and SoLA (Huang et al., 2025) require additional calibration data for compression, which introduces extra storage overhead and diminishes their practical utility.

To address this critical bottleneck, we propose a novel memory-efficient fine-tuning approach named **Lo**w-**R**ank **Act**ivation Compression (LORACT), which aims to reduce activation memory overhead while maintaining fine-tuning performance. By leveraging the inherent low-rank characteristic of model activation, LORACT employs an online compression mechanism during the forward pass to reduce the activation memory consumption, eliminating the requirement for any additional calibration data. To mitigate the computational overhead of the online compression, we introduce a novel sampling-based orthogonal decomposition algorithm that substantially enhances computational efficiency and decomposition quality. Furthermore, to facilitate the adaptability to the widely adopted Transformer-based foundation models, we design a tailored pre-norm activation compression strategy that ensures both compatibility and improved performance. To verify the effectiveness of LORACT, we conduct experiments on both language and vision tasks. The results show that LORACT consistently reduces activation memory while maintaining fine-tuning performance. Notably, by controlling the compression ratio, LORACT achieves competitive performance compared to LoRA while saving approximately 80% of the memory across all experiments.

The main contributions of this paper are summarized as follows:

- We empirically reveal the inherent low-rank nature of activations in the context of PEFT, thus validating the feasibility of activation compression.
- We propose LORACT, a flexible and versatile online activation compression strategy that can be applied to any differentiable functions and is free of any calibration data.
- To ensure computational efficiency, we introduce a novel sampling-based orthogonal decomposition algorithm that reduces computational overhead while maintaining theoretical guarantees.
- We design a tailored pre-norm compression mechanism that can be seamlessly integrated into Transformer-based models. The proposed method achieves approximately 80% of activation memory reduction compared to LoRA, while maintaining competitive performance.

## 2 BACKGROUND

### 2.1 IS PARAMETER-EFFICIENT EQUIVALENT TO MEMORY-EFFICIENT?

The parameter-efficient fine-tuning paradigm optimizes only a small subset of parameters while exhibiting remarkable performance, which has attracted considerable research interest (Houlsby et al., 2019; Hu et al., 2022; Ding et al., 2023; Han et al., 2024). The learning objective of parameter-efficient fine-tuning can be formulated as

$$L = \mathcal{L}\left(f(\boldsymbol{X}; \boldsymbol{\theta}, \boldsymbol{\theta}_0), \boldsymbol{Y}\right) \tag{1}$$

where $\boldsymbol{\theta}$ denotes the trainable parameters, $\boldsymbol{\theta}_0$ represents the fixed parameters, and $||\boldsymbol{\theta}||_0 \ll ||\boldsymbol{\theta}_0||_0$. During the backward pass, the trainable parameters $\boldsymbol{\theta}$ are updated along the negative gradient direction, i.e., $\nabla_{\boldsymbol{\theta}} L$, using the backpropagation algorithm (Rumelhart et al., 1986). However, the overall memory requirement is not solely determined by the number of trainable parameters. Specifically, memory consumption during fine-tuning arises mainly from the following four components:

- *Model parameters*, which comprises both trainable parameters $\boldsymbol{\theta}$ and fixed parameters $\boldsymbol{\theta}_0$. The memory usage is related to the quantity of model parameters and storage format.

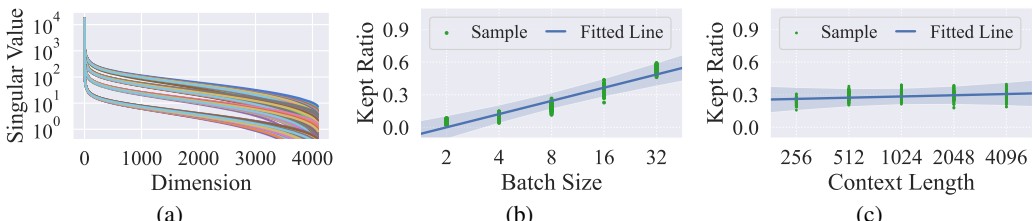

Figure 2: (a) The singular value distributions of model activations from different layers and batches. (b-c) The ratio of kept dimensions when 90% of energy is retained.

- *Gradients* of trainable parameters $g = \nabla_\theta L$, which are stored for parameter optimization. The memory consumption of gradients is equal to that of trainable parameters.

- *Optimizer states* for improving optimization, such as SGD momentum (Sutskever et al., 2013) and Adam (Kingma & Ba, 2015). The memory usage is also related to the trainable parameters.

- *Activations* for gradient computation, which are computed during the forward pass and retained for use in the backward pass. For example, in attention mechanism (Bahdanau et al., 2015), the inputs $Q$, $K$, $V$, and the attention score $\text{Softmax}(QK^\top/\sqrt{d})$ are stored to facilitate backpropagation. Similarly, for nonlinear functions such as $a = \text{Softmax}(x)$ and $a = \text{Sigmoid}(x)$, the output $a$ is preserved for computing the partial derivative $\partial a/\partial x$. In practice, the memory of activations is dependent on the model architecture.

In parameter-efficient fine-tuning, the number of trainable parameters is significantly smaller than the total number of model parameters. As a result, the memory required to store gradients and optimizer states is substantially reduced. However, the consumption of activation memory remains non-negligible. Although freezing model weights eliminates the need to store activations for the corresponding linear layers, a considerable amount of activation memory is still required for non-linear layers during backpropagation, such as the normalization layers, multi-head self-attention layers, and multi-layer perceptrons. Consequently, activation memory emerges as a dominant factor in overall memory consumption.

To evaluate the additional memory cost of fine-tuning, we perform a series of experiments on LLaMa2-7B (Touvron et al., 2023) using LoRA (Hu et al., 2022). The results are presented in Figure 1. Although the number of trainable parameters is substantially reduced, leading to a total memory cost of less than 1 GB for gradients and optimizer states, the activation memory remains considerably high. Moreover, the activation memory increases linearly with both batch size and context length. In some configurations, the memory demand can reach up to 64 GB, which even exceeds the total memory capacity of consumer-grade GPUs. Such considerable memory requirements make large batch sizes or context lengths infeasible in practical deployments.

## 2.2 Is Activation Memory Compressible?

The high activation memory overhead in parameter-efficient fine-tuning poses a significant challenge to its objective of achieving lightweight memory consumption. This naturally raises a question: *Can activation memory be effectively compressed?*

To explore this question, we first extract the activations from samples in the Alpaca dataset (Taori et al., 2023) using the LLaMa2-7B model (Touvron et al., 2023). We then analyze the singular values of these activation across different sample batches and model layers. The results illustrated in Figure 2a indicate that the singular values consistently exhibit a long-tailed distribution regardless of the source of activation. Notably, the largest singular value exceeds $10^4$, while most of the remaining values fall within the range of $10^0$ to $10^2$. This suggests that the activation matrices are essentially low-rank, with only a few dimensions capturing the majority of the information, whereas the rest contribute only marginally.

To further investigate the potential of activation compression under large batch sizes or context lengths, we further analyze the characteristics of activation with increased sizes. To quantify the low-rank nature of activations, we compute the ratio of kept dimensions when 90% of energy (i.e., the sum of singular values) is retained. The results are reported in Figures 2b and 2c. For small

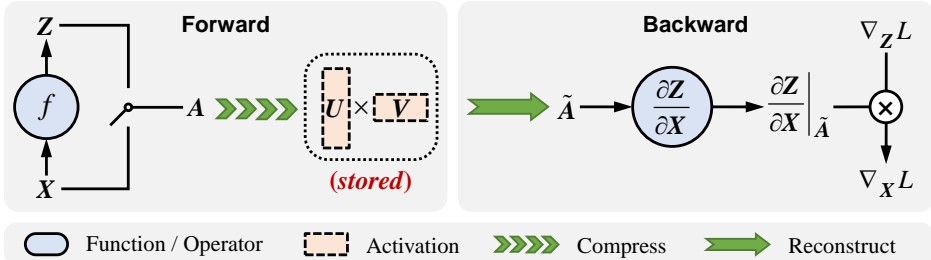

Figure 3: An illustration of LoRAct.

batch sizes of 2 or 4, only about 10% of the dimensions need to be retained to preserve 90% of the total energy. Even at a large batch size of 32, the kept ratio remains around 50%, which indicates significant potential for memory reduction via activation compression. Besides, increasing the context length has only a minor effect on the rank of activations, with the kept ratio stabilizing at approximately 30%, which further supports the feasibility of compressing activation sizes.

Some prior research has also explored the potential of activation compression, which however, is either unpractical or less accurate. For example, LoRA-FA (Zhang et al., 2023) freezes the down projection and optimizes only the up projection in each LoRA layer; however, it can only reduce limited size of activations. CompAct (Shamshoum et al., 2025) projects the activation of each linear layer into a low-rank subspace, which is incompatible with the prevailing parameter-efficient fine-tuning, where the activations from linear layers are already eliminated. CoLA (Liu et al., 2025) decomposes the full-size weight matrix into low-rank matrices, which inevitably harms the original inference capability. ESPACE (Sakr & Khailany, 2024) projects the activation using a static matrix, which loses flexibility when facing diverse activations. SoLA (Huang et al., 2025) conducts low-rank decomposition approach based on an additional calibration dataset, which is unfeasible in most practical scenarios. To overcome the limitations of existing works, we propose LoRAct, a flexible and versatile compressing method that facilitates memory-efficient fine-tuning.

## 3 PROPOSED METHOD

### 3.1 LoRAct: LOW-RANK ACTIVATION COMPRESSION

We propose a memory-efficient fine-tuning method termed *Low-Rank Activation Compression* (LoRAct). The overall framework is illustrated in Figure 3. Specifically, LoRAct first calculates the function output during the forward pass, and then extracts and decomposes the activation into low-rank approximations:

$$\boldsymbol{Z} = f(\boldsymbol{X}) \tag{2}$$

$$\boldsymbol{U}, \boldsymbol{V} = \text{Decompose}(\boldsymbol{A}, k) \tag{3}$$

where $\boldsymbol{A} \in \mathbb{R}^{m \times n}$ is the activation for computing $\partial \boldsymbol{Z} / \partial \boldsymbol{X}$, which typically comes from $\boldsymbol{X}$ or $\boldsymbol{Z}$. LoRAct applies low-rank matrix decomposition on $\boldsymbol{A}$ to obtain $\boldsymbol{A} \approx \boldsymbol{U}\boldsymbol{V}$, where $\boldsymbol{U} \in \mathbb{R}^{m \times k}$ and $\boldsymbol{V} \in \mathbb{R}^{k \times n}$ are two low-rank matrices. During the backward pass, LoRAct first reconstructs the activation by $\tilde{\boldsymbol{A}} = \boldsymbol{U}\boldsymbol{V}$, and then utilize $\tilde{\boldsymbol{A}}$ to calculate the partial derivative as well as the gradient:

$$\tilde{\boldsymbol{A}} = \text{Reconstruct}(\boldsymbol{U}, \boldsymbol{V}) \tag{4}$$

$$\nabla_{\boldsymbol{X}} L = \nabla_{\boldsymbol{Z}} L \cdot \left. \frac{\partial \boldsymbol{Z}}{\partial \boldsymbol{X}} \right|_{\tilde{\boldsymbol{A}}} \tag{5}$$

By compressing and storing only the low-rank matrices, LoRAct reduces the activation memory from $\mathcal{O}(mn)$ to $\mathcal{O}((m+n)k)$, where $k \ll m, n$. Furthermore, by incorporating an online compression mechanism, LoRAct adaptively compresses activations into their corresponding optimal components. This process is agnostic to the diverse properties of activations and eliminates the need for calibration data.

It is worth noting that LoRAct can be applied to any differentiable functions and modules. In contrast, existing activation compression methods only consider a single type of function, such as

compressing linear functions (Shamshoum et al., 2025) or normalization layers (Hu et al., 2025), which substantially limits their generalizability. Moreover, prior methods (Yu & Wu, 2023; Sakr & Khailany, 2024; Liu et al., 2025) compress activations at the expense of simplifying base modules or intermediate variables, which inevitably constrains the inherent capability of the base model. In contrast, LORACT does not compress any modules during the forward process, thus ensuring the basic inference performance of the model.

We prove that compressing activations during the forward pass, which is a common practice in many existing approaches, will lead to an accumulation of error with respect to the learning objective.

**Theorem 3.1.** *Suppose there are $N$ compressed activation matrices with the $i$-th compression incurring an error $\sigma_i$. Between consecutive activations there are fusion operators $\{f_i\}_{i=1}^{N-1}$, each being Lipschitz continuous with constant $L_i$. Let $L = \max_{i \in [N-1]} L_i$. For any input $\boldsymbol{x}$ and label $\boldsymbol{y}$, we have:*

$$\mathcal{L}(F(\boldsymbol{x}), \boldsymbol{y}) - \mathcal{L}(F_{comp}(\boldsymbol{x}), \boldsymbol{y}) \leq \sqrt{2} L_{\boldsymbol{W}} \Sigma_{i=1}^{N}(L^{N-i}\sigma_i)$$

*where $F$ denotes the overall forward mapping of the model, $L_{\boldsymbol{W}}$ is the Lipschitz constant of the final linear layer, $\mathcal{L}$ denotes the cross-entropy loss function.*

## 3.2 THE ENHANCED DECOMPOSITION ALGORITHM

LORACT employs an online matrix compression mechanism during the forward pass, and the effectiveness and efficiency of the algorithm are largely determined by this mechanism. In general, a matrix decomposition method approximates a large matrix $\boldsymbol{A}_{m \times n}$ by the product of two rank-$k$ matrices $\boldsymbol{U}_{m \times k}$ and $\boldsymbol{V}_{k \times n}$, thereby reducing the storage overhead.

Some previous work (Hao et al., 2024) attempts to approximate $\boldsymbol{A}$ by projecting its image into a lower-dimensional subspace via a random projection. Specifically, given a Gaussian matrix $\boldsymbol{G}_{l \times m}$, one defines $\boldsymbol{U} = \frac{1}{l}\boldsymbol{G}^{\top}$ and $\boldsymbol{V} = \boldsymbol{G}\boldsymbol{A}$, and approximates $\boldsymbol{A}$ by $\frac{1}{l}\boldsymbol{G}^{\top}\boldsymbol{G}\boldsymbol{A}$. However, this approximation can incur a large error and we provide a lower bound on its error.

**Theorem 3.2.** *Let $\boldsymbol{G}_{l \times m}$ be a standard Gaussian matrix. The expected deviation admits the approximate lower bound:*

$$\mathbb{E}\|\frac{1}{l}\boldsymbol{G}^{\top}\boldsymbol{G}\boldsymbol{A} - \boldsymbol{A}\| \geq \sqrt{\frac{m-l}{m+1}}\|\boldsymbol{A}\|$$

For better memory compression, it is typical to set $l \ll m$, in which case the lower bound is approximately $\|\boldsymbol{A}\|$. On the other hand, the error upper bound of Randomized SVD (RSVD) is $(1 + \sqrt{\frac{k}{l-k}})\sigma_{k+1}(\boldsymbol{A}) + \frac{e\sqrt{l}}{l-k}\sqrt{\sum_{j>k}\sigma_j^2}$ (Halko et al., 2011). Consequently, we establishe that RSVD outperforms random projection by a substantial margin, suggesting that approximate SVD methods constitute a more advantageous strategy.

*Remark* 4.1. *The conclusion we obtain is highly intuitive. This is because the RSVD method employs a Gaussian matrix to randomly approximate the column space of $\boldsymbol{A}$, whereas random projection instead approximates the entire space $\mathbb{R}^m$. Clearly, approximating the column space of $\boldsymbol{A}$ is an easier task, since $\boldsymbol{A}$ may be rank-deficient or even low-rank.*

While Truncated SVD provides the optimal rank-$k$ approximation (Eckart & Young, 1936), its computational cost is prohibitive at $\mathcal{O}(mn\min(m,n))$. RSVD reduce the complexity to $\mathcal{O}(mnk)$ and offer tight theoretical guarantees on approximation error, making them widely adopted in practice (Halko et al., 2011). However, in large-scale models, applying RSVD to compress activations still incurs substantial computational overhead, making it difficult to meet the efficiency requirements of LORACT. To overcome this limitation, we introduce a novel decomposition algorithm.

The computational cost of RSVD mainly arises from three components: generating the standard Gaussian matrix, performing the QR decomposition, and executing the matrix multiplications. Notably, the Nyström method (Drineas & Mahoney, 2005) does not require generating a Gaussian matrix, which motivates us to propose a Sampling-Based Orthogonal Decomposition algorithm that replaces the Gaussian matrix with a uniformly sampled submatrix $\boldsymbol{A}_k^{\top}$, while keeping the rest of the algorithm identical to RSVD. This design eliminates the need to generate a Gaussian matrix, thereby further reducing the computational overhead and combining the strengths of Nyström and RSVD. The detailed procedure of the algorithm is shown in Algorithm 1.

---

**Algorithm 1** SAMPLING-BASED ORTHOGONAL DECOMPOSITION

---

**Input:** Original matrix $A \in \mathbb{R}^{m \times n}$, expected rank $k$, number of iterations $t$.
**Output:** Decomposed matrices $U \in \mathbb{R}^{m \times k}$ and $V \in \mathbb{R}^{k \times n}$.
 1: Randomly sample $k$ rows of $A$, denoted as $A_k \in \mathbb{R}^{k \times n}$.
 2: Compute $Y = AA_k^\top$.
 3: **for** $i = 1, \cdots, t$ **do**
 4:     Compute QR decomposition of $Y = QR$, where $Q \in \mathbb{R}^{m \times k}$.
 5:     Compute $Y = A(A^\top Q)$.
 6: **end for**
 7: Compute QR decomposition of $Y = QR$, where $Q \in \mathbb{R}^{m \times k}$.
 8: Form $U = Q$ and compute $V = Q^\top A$.

---

To establish the effectiveness of our algorithm, we provide a theoretical upper bound on its approximation error. Similar to RSVD, our method employs a test matrix to perform random sampling on the column space of $A$, thereby approximately preserving it in a lower-dimensional subspace. Therefore, the classical result established by Halko et al. (2011) in Theorem 3.3 is highly relevant, for it underpins all fundamental results on SVD approximation error using test matrices.

**Theorem 3.3.** *Suppose that* $A = U\Sigma V^\top = U \begin{bmatrix} \Sigma_1 & 0 \\ 0 & \Sigma_2 \\ 0 & 0 \end{bmatrix} \begin{bmatrix} V_1^\top \\ V_2^\top \end{bmatrix}$ *and* $\Omega$ *is an arbitrary test matrix, where* $V_1$ *is the submatrix consisting of the first* $k$ *columns of* $V$ *and* $V_2$ *is the submatrix consisting of the remaining* $n - k$ *columns of* $V$. $\Sigma_2$ *is the the diagonal matrix containing the last* $n - k$ *singular values of* $A$. *We define that* $Y = A\Omega$, $\Omega_1 = V_1^\top \Omega$ *and* $\Omega_2 = V_2^\top \Omega$.

*If* $\Omega_1$ *has full row rank, we have:* $||A - P_Y A||^2 \leq ||\Sigma_2||^2 + ||\Sigma_2 \Omega_2 \Omega_1^+||^2$, *where* $\Omega_1^+$ *denotes the Moore-Penrose pseudoinverse of* $\Omega_1$, $P_Y$ *denotes the orthogonal projector onto* $\mathrm{range}(Y)$.

Theorem 3.3 gives a deterministic bound for the SVD approximation error using test matrix. The proof of this theorem in Halko et al. (2011) contains certain inaccuracies; hence, we restate the theorem here and present a corrected proof in the appendix. Based on this, we further derive the theoretical upper bound of the approximation error for our proposed method.

**Theorem 3.4.** *For a matrix* $A_{m \times n}$, *suppose that we perform uniform random row sampling to obtain a submatrix* $A_l \in \mathbb{R}^{l \times n}$, *and let the test matrix* $\Omega := A_l^\top$. *Then we have the expectation bound*

$$\mathbb{E}||A - QQ^T A|| \leq (1 + C\sqrt{\frac{\mu_k k}{l}})\sigma_{k+1}(A) + ke^{\frac{1}{c}}||A||,$$

*where* $C$ *is constant and* $\mu_k := \frac{m}{k} \max_{1 \leq i \leq m} ||(U_k)_{i,:}||_2$. $U_k$ *denotes the first* $k$ *columns of* $U$.

In summary, Theorem 3.4 shows that the error upper bound of our sampling-based method is relatively tight, explicitly depending on the $k$-coherence $\mu_k$ of $A$ through the subspace structure of $U_k$. This result indicates that our approach preserves the comparable theoretical approximation guarantees as RSVD. At the same time, by avoiding the generation of Gaussian test matrices, our method achieves reduced computational overhead without incurring additional approximation error, which meets our requirement. All the theorems are proved in detail in the Appendix.

### 3.3 ADAPTATION TO TRANSFORMER-BASED MODELS

Transformer-based models (Vaswani et al., 2017; Dosovitskiy et al., 2021) have been widely adopted in various scenarios, and serve as a fundamental architecture for many large-scale foundation models (Touvron et al., 2023; Liu et al., 2024). Therefore, it is necessary to adapt LoRAct to the prevailing Transformer-based models. One potential solution is to treat each Transformer layer as a black-box and directly compress the activation of input of each layer. However, such approach neglects the intrinsic architecture inside the layer, and may lead to redundant activation memory and unexpected estimation error.

To overcome this challenge, we introduce a *pre-norm activation compression* strategy, which is tailored for the fundamental pre-norm unit in each Transformer layer. Formally, a Transformer layer

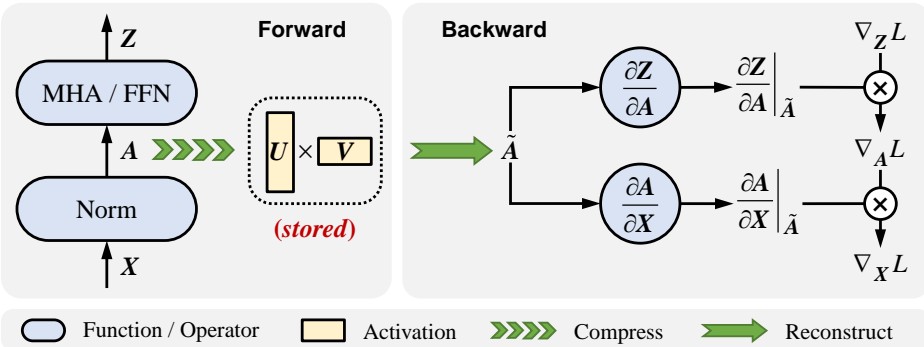

Figure 4: The adaptation of LORACT for Transformer-based models. LORACT leverages the convenience of the pre-norm structure to further save memory and computational costs.

can be expressed as

$$\boldsymbol{Z} = \mathcal{F}\left(\mathrm{Norm}\left(\boldsymbol{X}\right)\right) + \boldsymbol{X} \tag{6}$$

$$\mathrm{Norm}\left(\boldsymbol{X}\right) = \frac{\boldsymbol{X}}{\mathrm{RMS}(\boldsymbol{X})} \cdot \boldsymbol{\gamma} \tag{7}$$

where $\mathcal{F}$ represents a sub-layer, which can be either a Multi-Head Attention (MHA) or a Feed-Forward Network (FFN). RMS denotes root mean square (Zhang & Sennrich, 2019), and $\boldsymbol{\gamma}$ is a scaling vector. This structure is referred to as a pre-norm unit (Wang et al., 2019). In conventional fine-tuning frameworks, both Norm and $\mathcal{F}$ store their inputs as activations for gradient computation, which can lead to additional memory overhead. To mitigate this issue, we propose to store the output of normalization $\boldsymbol{A} = \mathrm{Norm}(\boldsymbol{X})$ as the activation, and recover the partial derivatives as:

$$\bar{\boldsymbol{X}} = \boldsymbol{A}/\boldsymbol{\gamma} \tag{8}$$

$$\nabla_{\bar{\boldsymbol{X}}} L = \nabla_{\boldsymbol{A}} L \cdot \boldsymbol{\gamma} \tag{9}$$

$$\nabla_{\boldsymbol{X}} L = \frac{1}{\mathrm{RMS}} \left( \nabla_{\bar{\boldsymbol{X}}} L - \bar{\boldsymbol{X}} \cdot \frac{1}{n} \sum_{j=1}^{n} \left[ \nabla_{\bar{\boldsymbol{X}}} L \cdot \bar{\boldsymbol{X}} \right]_{\cdot,j} \right) \tag{10}$$

This strategy allows the Norm function and the sub-layer $\mathcal{F}$ to share the same activation memory, since the output of Norm is exactly the input of $\mathcal{F}$. Note that the RMS value should also be stored for computing $\nabla_{\boldsymbol{X}} L$, but its size is relatively small (a one-dimensional vector) and the memory cost is negligible. Moreover, by applying a structure-agnostic mechanism for the sub-layer $\mathcal{F}$, we equip LORACT with the flexibility to adapt to complex sub-layers, which may undergo frequent updates with the evolvement of latest foundation models.

## 4 EXPERIMENTS

### 4.1 EXPERIMENTAL SETTING

To assess the generalizability of LORACT, we evaluate its performance on both language and vision tasks. For language task, we consider instruction tuning and language modeling problems. For instruction tuning problem, we fine-tune models on the Alpaca (Taori et al., 2023) and FLAN-v2 (Longpre et al., 2023) instruction datasets. Following prior work (Dettmers et al., 2023), we subsample 50k training examples from FLAN-v2 to match the scale of Alpaca. The fine-tuned models are evaluated on the MMLU benchmark (Hendrycks et al., 2021) using the average 5-shot metric, following the settings of prior works (Dettmers et al., 2023). For language modeling problem, we fine-tune model on the WikiText-2 dataset (Merity et al., 2017) and evaluate the performance using the perplexity metric.

For vision task, we evaluate on image classification using two datasets, including CIFAR-100 (Krizhevsky et al., 2009) and Food-101 (Bossard et al., 2014), and calculate the top-1 accuracy on their corresponding test datasets.

Table 1: Evaluation scores and memory usage on Alpaca and FLAN-v2.

| Methods | Alpaca | | | FLAN-v2 | | |
|---------|---------|---------|---------|---------|---------|---------|
| | Score (↑) | Total Mem. | Act. Mem. | Score (↑) | Total Mem. | Act. Mem. |
| Base model | 45.89 | 12.465G | - | 45.89 | 12.465G | - |
| LoRA | 46.18 (+0.29) | 29.748G | 16.847G | 46.04 (+0.15) | 29.748G | 16.847G |
| LoRAct ($r = 1/2$) | 46.29 (+0.40) | 15.584G | 2.712G | 45.71 (-0.18) | 15.584G | 2.712G |
| LoRAct ($r = 1/4$) | 46.41 (+0.52) | 14.599G | 1.728G | 45.95 (+0.06) | 14.599G | 1.728G |
| LoRAct ($r = 1/8$) | 46.44 (+0.55) | 14.110G | 1.238G | 45.90 (+0.01) | 14.110G | 1.238G |
| LoRAct ($r = 1/16$) | 46.60 (+0.71) | 13.861G | 0.989G | 45.76 (-0.13) | 13.861G | 0.989G |
| LoRAct ($r = 1/32$) | 46.31 (+0.42) | 13.738G | 0.866G | 46.20 (+0.31) | 13.738G | 0.866G |
| LoRAct ($r = 1/64$) | 46.29 (+0.40) | 13.676G | 0.805G | 45.86 (-0.03) | 13.676G | 0.805G |

Table 2: Perplexity and memory usage on WikiText-2.

| Methods | Perplexity (↓) | Total Memory | Activation Memory |
|---------|---------------|--------------|-------------------|
| Base model | 5.11 | 12.323G | - |
| LoRA | 4.84 (-0.27) | 19.967G | 7.566G |
| LoRAct ($r = 1/2$) | 4.93 (-0.18) | 13.441G | 1.465G |
| LoRAct ($r = 1/4$) | 4.96 (-0.15) | 13.172G | 1.056G |
| LoRAct ($r = 1/8$) | 4.98 (-0.13) | 13.047G | 0.786G |
| LoRAct ($r = 1/16$) | 5.20 (+0.09) | 12.984G | 0.662G |

Table 3: Comparison of layer-wise and pre-norm activation compression on Alpaca and FLAN-v2.

| Datasets | Methods | $r = 1/2$ | $r = 1/4$ | $r = 1/8$ | $r = 1/16$ | $r = 1/32$ | $r = 1/64$ |
|----------|---------|-----------|-----------|-----------|------------|------------|------------|
| Alpaca | layer-wise | 46.24 | 46.27 | 46.30 | 46.33 | 46.25 | 46.16 |
| | pre-norm | **46.29** | **46.41** | **46.44** | **46.60** | **46.31** | **46.29** |
| FLAN-v2 | layer-wise | 45.32 | 45.90 | 45.85 | 45.04 | 45.39 | 45.27 |
| | pre-norm | **45.71** | **45.95** | **45.90** | **45.76** | **46.20** | **45.86** |

Table 4: Comparison of layer-wise and pre-norm activation compression on WikiText-2.

| Datasets | Methods | $r = 1/2$ | $r = 1/4$ | $r = 1/8$ | $r = 1/16$ |
|----------|---------|-----------|-----------|-----------|------------|
| WikiText-2 | layer-wise | 4.97 | 4.98 | 5.01 | 5.97 |
| | pre-norm | **4.93** | **4.96** | **4.98** | **5.20** |

We fine-tune the LLaMa2-7B model (Touvron et al., 2023) for language tasks. The learning rate is set to $3 \times 10^{-5}$ with a batch size of 128, and a micro batch size of 16. The context length is set to 256 for Alpaca and FLAN-v2, and 128 for WikiText-2. For vision tasks, we fine-tune the ViT-B/16 model (Dosovitskiy et al., 2021) with a learning rate of 0.1, a batch size of 2048 and a micro-batch size of 1024. For all experiments, we employ LoRA (Hu et al., 2022) modules with a bottleneck dimension of 64 and perform fine-tuning in BF16 precision. We apply gradient checkpointing to the sub-layers of each Transformer layer to automatically compute their gradients by storing only the inputs. All experiments are conducted using a single GPU with 32GB of memory.

## 4.2 INSTRUCTION TUNING AND LANGUAGE MODELING

We first evaluate the effectiveness of LoRAct when adapted to language models. Specifically, we experiment on the instruction tuning task using Alpaca and FLAN-v2 datasets, and compare the base model, LoRA, and LoRAct under varying compression ratios ($r = k/n$). Table 1 reports the evaluation metrics along with total and activation memory consumption. Across all settings, LoRAct can substantially reduce the memory usage while preserving the fine-tuning performance. Particularly, the activation memory falls below 1G when setting $r$ to less than $1/16$. Notably, smaller compression ratio $r$ can even yield better performance, such as $r = 1/16$ on Alpaca and $r = 1/32$ on FLAN-v2. This indicates that compressing activations by discarding small singular values may even facilitate the activation quality.

Table 5: Accuracy and memory usage on CIFAR-100 and Food-101.

| Methods | CIFAR-100 | | | Food-101 | | |
|---|---|---|---|---|---|---|
| | Score ($\uparrow$) | Total Mem. | Act. Mem. | Score ($\uparrow$) | Total Mem. | Act. Mem. |
| LoRA | 88.94 | 28.275G | 28.080G | 82.54 | 28.275G | 28.080G |
| LoRAct ($r = 1/2$) | 88.27 (-0.67) | 5.104G | 4.765G | 81.68 (-0.86) | 5.104G | 4.765G |
| LoRAct ($r = 1/4$) | 87.10 (-1.84) | 4.230G | 3.891G | 81.52 (-1.02) | 4.230G | 3.891G |
| LoRAct ($r = 1/8$) | 81.03 (-7.91) | 3.793G | 3.454G | 78.80 (-3.74) | 3.793G | 3.454G |
| LoRAct ($r = 1/16$) | 75.97 (-12.97) | 3.575G | 3.236G | 64.16 (-18.38) | 3.575G | 3.236G |

Table 6: Comparison of layer-wise and pre-norm activation compression on two image datasets.

| Methods | CIFAR-100 | | | | Food-101 | | | |
|---|---|---|---|---|---|---|---|---|
| | 1/2 | 1/4 | 1/8 | 1/16 | 1/2 | 1/4 | 1/8 | 1/16 |
| layer-wise | 88.01 | 84.12 | 61.19 | 7.79 | 81.05 | 80.55 | 75.42 | 4.13 |
| pre-norm | **88.27** | **87.10** | **81.03** | **75.97** | **81.68** | **81.52** | **78.80** | **64.16** |

Additionally, we conduct experiments on the language modeling task and report the results in Table 2. Specially, we use the WikiText-2 dataset and compare the base model, LoRA and LoRAct under varying compression ratios ($r = k/n$). The results show that LoRAct attains performance comparable to LoRA fine-tuning while reduing the activation memory usage. It is worth noting that as $r$ increases, the corresponding perplexity generally decreases, which indicates that retaining more activation information benefits language modeling performance.

Moreover, we compare the proposed pre-norm activation compression with the layer-wise compression scheme, and present the results in Tables 3 and 4. Across all settings, the pre-norm activation compression strategy consistently achieves higher score. We attribute this to greater error accumulation under layer-wise activation compression, whereas pre-norm activation compression mitigates error propagation and yields more accurate gradients during the backward pass.

### 4.3 Image Classification

To assess the generalizability of LoRAct to vision tasks, we conduct experiments on image classification tasks including CIFAR-100 and Food-101, and report the results in Table 5. The results show that LoRAct substantially reduces memory usage across all settings. When setting $r = 1/2$ or $1/4$, LoRAct can achieve performance comparable to LoRA on both the CIFAR-100 and Food-101 datasets. Notably, when $r < 1/8$, accuracy tends to decline, suggesting that, for vision tasks, LoRAct benefits from relatively larger $r$ to preserve sufficient information.

Moreover, we compare the performance of LoRAct using pre-norm activation compression against the layer-wise compression approach. The experimental results are reported in Table 6. Consistent with the language task, the pre-norm activation compression strategy outperforms the layer-wise compression strategy across all settings. These results further corroborate the effectiveness of the proposed pre-norm activation compression compared with the layer-wise compression strategy.

## 5 Conclusion

In this paper, we propose a novel memory-efficient fine-tuning approach *Low-Rank Activation Compression* (LoRAct), which is motivated by the low-rank nature of activations. LoRAct introduces a flexible and versatile compression strategy that can be applied online during the forward pass, eliminating the need for calibration data. Furthermore, LoRAct incorporates a novel sampling-based orthogonal decomposition tailored to low-rank matrices, yielding greater computational efficiency and tighter error bounds. Extensive experiments demonstrate that LoRAct can achieve competitive fine-tuning performance across both language and vision tasks while reducing the activation memory usage by approximately 80% compared to the broadly adopted LoRA method. We hope our method can assist the community in exploring the potential of memory-efficient fine-tuning.

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

## A APPENDIX

### A.1 PROOF OF THEOREM 3.1

We assume that in the normal case the activation matrices are denoted by $\boldsymbol{A}_1, \boldsymbol{A}_2, \ldots \boldsymbol{A}_N$, while in the forward-compression case the corresponding activation matrices are denoted by $\boldsymbol{A}_1^{'}, \boldsymbol{A}_2^{'}, \ldots \boldsymbol{A}_N^{'}$. By the given condition, we obtain $||\boldsymbol{A}_1 - \boldsymbol{A}_1^{'}|| \le \sigma_1$.

Invoking the Lipschitz continuity of $f_1$, it follows that

$$||f_1(\boldsymbol{A}_1) - f_1(\boldsymbol{A}_1^{'})|| \le L||\boldsymbol{A}_1 - \boldsymbol{A}_1^{'}|| \le L\sigma_1$$

In the normal case, we have $\boldsymbol{A}_2 = f_1(\boldsymbol{A}_1)$. While in the forward-compression case, we further compress $f_1(\boldsymbol{A}_1^{'})$ to obtain $\boldsymbol{A}_2^{'}$. Similarly, $||\boldsymbol{A}_2^{'} - f_1(\boldsymbol{A}_1^{'})|| \le \sigma_2$.

By the triangle inequality of the norm, we obtain:

$$||\boldsymbol{A}_2 - \boldsymbol{A}_2^{'}|| \le L\sigma_1 + \sigma_2$$

Proceeding in the same manner, we obtain:

$$||\boldsymbol{A}_N - \boldsymbol{A}_N^{'}|| \le \Sigma_{i=1}^{N}(L^{N-i}\sigma_i)$$

Subsequently, the final activation is passed through the model's linear layer, which is necessarily a Lipschitz continuous function. We assume its Lipschitz constant is denoted by $L_{\boldsymbol{W}}$.

Moreover, since the cross-entropy loss is globally Lipschitz continuous with respect to the logits, with Lipschitz constant $\sqrt{2}$.

Thus, we obtain the desired result:

$$\mathcal{L}(F(\boldsymbol{x}), \boldsymbol{y}) - \mathcal{L}(F_{comp}(\boldsymbol{x}), \boldsymbol{y}) \le \sqrt{2}L_{\boldsymbol{W}}\Sigma_{i=1}^{N}(L^{N-i}\sigma_i)$$

□

### A.2 PROOF OF THEOREM 3.2

Since $\boldsymbol{G}$ is a standard Gaussian matrix, it is with high probability of full row rank. As our goal is to derive a lower bound for $\mathbb{E}||\frac{1}{l}\boldsymbol{G}^T\boldsymbol{G}\boldsymbol{A} - \boldsymbol{A}||$, we may assume that $\boldsymbol{G}$ is indeed of full row rank. In this case, $\text{rank}(\boldsymbol{G}) = m - l = k$.

We denote by $N = \text{Null}(\boldsymbol{G})$ the null space of $\boldsymbol{G}$, with $\dim(N) = k$. Let $\boldsymbol{P}_N$ denote the orthogonal projection onto $N$. Then $\forall \boldsymbol{x} \in \mathbb{R}^m$, we have:

$$||(\frac{1}{l}\boldsymbol{G}^T\boldsymbol{G}\boldsymbol{A} - \boldsymbol{A})\boldsymbol{x}||^2 = ||(\frac{1}{l}\boldsymbol{G}^T\boldsymbol{G} - \boldsymbol{I})\boldsymbol{A}\boldsymbol{x}||^2$$

$$= ||(\frac{1}{l}\boldsymbol{G}^T\boldsymbol{G} - \boldsymbol{I})\boldsymbol{P}_N\boldsymbol{A}\boldsymbol{x}||^2 + ||(\frac{1}{l}\boldsymbol{G}^T\boldsymbol{G} - \boldsymbol{I})(\boldsymbol{I} - \boldsymbol{P}_N)\boldsymbol{A}\boldsymbol{x}||^2$$

$$\ge ||(\frac{1}{l}\boldsymbol{G}^T\boldsymbol{G} - \boldsymbol{I})\boldsymbol{P}_N\boldsymbol{A}\boldsymbol{x}||^2 = ||\boldsymbol{P}_N\boldsymbol{A}\boldsymbol{x}||^2$$

Since this inequality holds for all $\boldsymbol{x}$, we may take the supremum over both sides to obtain:

$$||(\frac{1}{l}\boldsymbol{G}^T\boldsymbol{G}\boldsymbol{A} - \boldsymbol{A})|| \ge ||\boldsymbol{P}_N\boldsymbol{A}||$$

Let $\boldsymbol{v}_1$, $\boldsymbol{u}_1$, and $\sigma_1$ denote the first left singular vector, the first right singular vector, and the largest singular value of $\boldsymbol{A}$, respectively. we have:

$$||\boldsymbol{P}_N\boldsymbol{A}|| = \max_{||\boldsymbol{x}||=1} ||\boldsymbol{P}_N\boldsymbol{A}\boldsymbol{x}|| \ge ||\boldsymbol{P}_N\boldsymbol{A}\boldsymbol{v}_1|| = ||\boldsymbol{P}_N\sigma_1\boldsymbol{u}_1|| = ||\boldsymbol{A}||||\boldsymbol{P}_N\boldsymbol{u}_1||$$

Therefore, $\mathbb{E}||\boldsymbol{P}_N\boldsymbol{A}|| \ge \mathbb{E}||\boldsymbol{A}||||\boldsymbol{P}_N\boldsymbol{u}_1|| = ||\boldsymbol{A}||\mathbb{E}||\boldsymbol{P}_N\boldsymbol{u}_1||$. It thus remains to analyze $\mathbb{E}||\boldsymbol{P}_N\boldsymbol{u}_1||$.

Because $\boldsymbol{G}$ is a standard Gaussian matrix independent of $\boldsymbol{A}$, $N$ is a uniformly random $k$-dimensional subspace of $\mathbb{R}^m$ on the Grassmann manifold. For any fixed unit vector $\boldsymbol{u}_1$, the random variable

$||\boldsymbol{P}_N \boldsymbol{u}_1||^2$ has the same distribution as the squared length obtained by projecting a uniformly distributed vector on the unit sphere $S^{m-1}$ onto a random $k$-dimensional subspace. A classical result states that:

$$||\boldsymbol{P}_N \boldsymbol{u}_1||^2 \sim \text{Beta}(\frac{k}{2}, \frac{m-k}{2})$$

Therefore, $\mathbb{E}||\boldsymbol{P}_N \boldsymbol{u}_1|| = \frac{\Gamma(\frac{k+1}{2})\Gamma(\frac{m}{2})}{\Gamma(\frac{k}{2})\Gamma(\frac{m+1}{2})} \geq \sqrt{\frac{m-l}{m+1}}$. So $\mathbb{E}||\boldsymbol{P}_N \boldsymbol{A}|| \geq \sqrt{\frac{m-l}{m+1}}\mathbb{E}||\boldsymbol{A}||$. Hence, we have established the conclusion of Theorem 3.2. $\square$

### A.3    PROOF OF THEOREM 3.3

For ease of exposition, we first introduce some notation:

$$\boldsymbol{A} = \boldsymbol{U}\boldsymbol{\Sigma}\boldsymbol{V}^T = \boldsymbol{U} \begin{bmatrix} \boldsymbol{\Sigma}_1 & \boldsymbol{0} \\ \boldsymbol{0} & \boldsymbol{\Sigma}_2 \\ \boldsymbol{0} & \boldsymbol{0} \end{bmatrix} \begin{bmatrix} \boldsymbol{V}_1^T \\ \boldsymbol{V}_2^T \end{bmatrix}$$

$$\boldsymbol{Y} = \boldsymbol{A}\boldsymbol{\Omega} = \boldsymbol{Q}\boldsymbol{R}$$

We denote that $\boldsymbol{\Omega}_1 = \boldsymbol{V}_1^T\boldsymbol{\Omega}, \boldsymbol{\Omega}_2 = \boldsymbol{V}_2^T\boldsymbol{\Omega}$, then

$$\boldsymbol{Y} = \boldsymbol{U}\boldsymbol{\Sigma} \begin{bmatrix} \boldsymbol{\Omega}_1 \\ \boldsymbol{\Omega}_2 \end{bmatrix} = \boldsymbol{U} \begin{bmatrix} \boldsymbol{\Sigma}_1\boldsymbol{\Omega}_1 \\ \boldsymbol{\Sigma}_2\boldsymbol{\Omega}_2 \\ \boldsymbol{0} \end{bmatrix}$$

*Proof.* We now begin the proof. We denote that $\boldsymbol{A}_r = \boldsymbol{U}^T\boldsymbol{A} = \begin{bmatrix} \boldsymbol{\Sigma}_1\boldsymbol{V}_1 \\ \boldsymbol{\Sigma}_2\boldsymbol{V}_2 \\ \boldsymbol{0} \end{bmatrix}, \boldsymbol{Y}_r = \boldsymbol{A}_r\boldsymbol{\Omega} = \begin{bmatrix} \boldsymbol{\Sigma}_1\boldsymbol{\Omega}_1 \\ \boldsymbol{\Sigma}_2\boldsymbol{\Omega}_2 \\ \boldsymbol{0} \end{bmatrix}$.

Then it can be derived

$$||\boldsymbol{A} - \boldsymbol{Q}\boldsymbol{Q}^T\boldsymbol{A}|| = ||(\boldsymbol{I} - \boldsymbol{P}_Y)\boldsymbol{A}|| = ||\boldsymbol{U}^T(\boldsymbol{I} - \boldsymbol{P}_Y)\boldsymbol{U}\boldsymbol{A}_r|| = ||(\boldsymbol{I} - \boldsymbol{P}_{Y_r})\boldsymbol{A}_r||$$

where $\boldsymbol{P}_Y$ represents an orthogonal projection operator that satisfies $\text{Range}(\boldsymbol{P}_Y) = \text{Range}(\boldsymbol{Y})$.

Firstly, we consider the simpler condition where $\text{rank}(\boldsymbol{A}) \leq k$. At this point, it is evident that $\boldsymbol{\Sigma}_2 = \boldsymbol{0}$, thus we can observe:

$$\boldsymbol{A}_r = \begin{bmatrix} \boldsymbol{\Sigma}_1\boldsymbol{V}_1 \\ \boldsymbol{0} \\ \boldsymbol{0} \end{bmatrix} \quad \text{and} \quad \boldsymbol{Y}_r = \begin{bmatrix} \boldsymbol{\Sigma}_1\boldsymbol{\Omega}_1 \\ \boldsymbol{0} \\ \boldsymbol{0} \end{bmatrix}$$

Since $\boldsymbol{\Omega}_1$ is of full row rank, it is evident that $\text{Range}(\boldsymbol{\Sigma}_1\boldsymbol{V}_1) = \text{Range}(\boldsymbol{\Sigma}_1\boldsymbol{\Omega}_1)$. That is $\text{Range}(\boldsymbol{A}_r) = \text{Range}(\boldsymbol{Y}_r)$. Thus, $\text{Range}(\boldsymbol{A}_r) = \text{Range}(\boldsymbol{P}_{Y_r})$. As $\boldsymbol{P}_{Y_r}$ is the orthogonal projector onto $\text{Range}(\boldsymbol{A}_r)$, for any vector $\boldsymbol{x}$, $\boldsymbol{P}_{Y_r}\boldsymbol{A}_r\boldsymbol{x} = \boldsymbol{A}_r\boldsymbol{x}$. Thus, we conclude that $\boldsymbol{P}_{Y_r}\boldsymbol{A}_r = \boldsymbol{A}_r$ and thereby establish the conclusion of the theorem 3.3:

$$||\boldsymbol{A} - \boldsymbol{Q}\boldsymbol{Q}^T\boldsymbol{A}|| = ||(\boldsymbol{I} - \boldsymbol{P}_{Y_r})\boldsymbol{A}_r|| = 0$$

In the following, we address the more involved case where $\text{rank}(\boldsymbol{A}) > k$. The idea of the proof is to reformulate $\boldsymbol{P}_{Y_r}$ as a matrix with a more tractable structure, thereby facilitating a bound on $||\boldsymbol{A} - \boldsymbol{Q}\boldsymbol{Q}^T\boldsymbol{A}||$. Since $\text{rank}(\boldsymbol{A}) > k$, the matrix $\boldsymbol{\Sigma}_1$ is $k$-rank full, and as a result, $\boldsymbol{\Sigma}_1\boldsymbol{\Omega}_1$ is of full row rank. Thus, we can reformulate $\boldsymbol{P}_{Y_r}$ as a block matrix, with one block being the identity matrix $\boldsymbol{I}_k$, while the remaining part is considered as an interference term, which allows us to perform the scaling. Specifically: let

$$\boldsymbol{Z} = \boldsymbol{Y}_r \cdot \boldsymbol{\Omega}_1^+\boldsymbol{\Sigma}_1^{-1} = \begin{bmatrix} \boldsymbol{I} \\ \boldsymbol{F} \\ \boldsymbol{0} \end{bmatrix}, \text{where } \boldsymbol{F} = \boldsymbol{\Sigma}_2\boldsymbol{\Omega}_2\boldsymbol{\Omega}_1^+\boldsymbol{\Sigma}_1^{-1}$$

It follows immediately that $\text{Range}(\boldsymbol{Z}) \subseteq \text{Range}(\boldsymbol{Y}_r)$, Consequently, $\text{Range}(\boldsymbol{P}_Z) \subseteq \text{Range}(\boldsymbol{P}_{Y_r})$. Hence, we derive the bound:

$$||(\boldsymbol{I} - \boldsymbol{P}_{Y_r})\boldsymbol{A}_r|| \leq ||(\boldsymbol{I} - \boldsymbol{P}_Z)\boldsymbol{A}_r||$$

We know that $||\boldsymbol{A}||$ equals the largest singular value of $\boldsymbol{A}$, also the square root of the largest eigenvalue of $\boldsymbol{A}^T\boldsymbol{A}$, we can further bound it as follows:

$$||(\boldsymbol{I} - \boldsymbol{P_Z})\boldsymbol{A}_r||^2 = \lambda(\boldsymbol{A}_r^T(\boldsymbol{I} - \boldsymbol{P_Z})^T(\boldsymbol{I} - \boldsymbol{P_Z})\boldsymbol{A}_r) = ||\boldsymbol{A}_r^T(\boldsymbol{I} - \boldsymbol{P_Z})\boldsymbol{A}_r|| = ||\boldsymbol{\Sigma}^T(\boldsymbol{I} - \boldsymbol{P_Z})\boldsymbol{\Sigma}||$$

The identities follow from the fact that an orthogonal projection operator is always idempotent, and the eigenvalues of a positive semi-definite matrix coincide with its singular values.

As $\boldsymbol{Z}$ has full column rank, it follows directly from the properties of orthogonal projection operators that:

$$\boldsymbol{P_Z} = \boldsymbol{Z}(\boldsymbol{Z}^T\boldsymbol{Z})^{-1}\boldsymbol{Z}^T = \begin{bmatrix} (\boldsymbol{I} + \boldsymbol{F}^T\boldsymbol{F})^{-1} & (\boldsymbol{I} + \boldsymbol{F}^T\boldsymbol{F})^{-1}\boldsymbol{F}^T & \boldsymbol{0} \\ \boldsymbol{F}(\boldsymbol{I} + \boldsymbol{F}^T\boldsymbol{F})^{-1} & \boldsymbol{F}(\boldsymbol{I} + \boldsymbol{F}^T\boldsymbol{F})^{-1}\boldsymbol{F}^T & \boldsymbol{0} \\ \boldsymbol{0} & \boldsymbol{0} & \boldsymbol{0} \end{bmatrix}$$

Therefore,

$$\boldsymbol{I} - \boldsymbol{P_Z} = \begin{bmatrix} \boldsymbol{I} - (\boldsymbol{I} + \boldsymbol{F}^T\boldsymbol{F})^{-1} & -(\boldsymbol{I} + \boldsymbol{F}^T\boldsymbol{F})^{-1}\boldsymbol{F}^T & \boldsymbol{0} \\ -\boldsymbol{F}(\boldsymbol{I} + \boldsymbol{F}^T\boldsymbol{F})^{-1} & \boldsymbol{I} - \boldsymbol{F}(\boldsymbol{I} + \boldsymbol{F}^T\boldsymbol{F})^{-1}\boldsymbol{F}^T & \boldsymbol{0} \\ \boldsymbol{0} & \boldsymbol{0} & \boldsymbol{I} \end{bmatrix}$$

According to Halko et al. (2011), for any positive semidefinite matrix $\boldsymbol{M} \succeq \boldsymbol{0}$, we have $\boldsymbol{I} - (\boldsymbol{I} + \boldsymbol{M})^{-1} \preceq \boldsymbol{M}$. Clearly $\boldsymbol{F}^T\boldsymbol{F} \succeq \boldsymbol{0}$, hence, $\boldsymbol{I} - (\boldsymbol{I} + \boldsymbol{F}^T\boldsymbol{F})^{-1} \preceq \boldsymbol{F}^T\boldsymbol{F}$. Moreover, since $\boldsymbol{I} - \boldsymbol{F}(\boldsymbol{I} + \boldsymbol{F}^T\boldsymbol{F})^{-1}\boldsymbol{F}^T \preceq \boldsymbol{I}$, we can obtain the following strong conclusion:

$$\boldsymbol{I} - \boldsymbol{P_Z} \preceq \begin{bmatrix} \boldsymbol{F}^T\boldsymbol{F} & -(\boldsymbol{I} + \boldsymbol{F}^T\boldsymbol{F})^{-1}\boldsymbol{F}^T & \boldsymbol{0} \\ -\boldsymbol{F}(\boldsymbol{I} + \boldsymbol{F}^T\boldsymbol{F})^{-1} & \boldsymbol{I} & \boldsymbol{0} \\ \boldsymbol{0} & \boldsymbol{0} & \boldsymbol{I} \end{bmatrix}$$

Thus,

$$\boldsymbol{\Sigma}^T(\boldsymbol{I} - \boldsymbol{P_Z})\boldsymbol{\Sigma} \preceq \begin{bmatrix} \boldsymbol{\Sigma}_1\boldsymbol{F}^T\boldsymbol{F}\boldsymbol{\Sigma}_1 & -\boldsymbol{\Sigma}_1(\boldsymbol{I} + \boldsymbol{F}^T\boldsymbol{F})^{-1}\boldsymbol{F}^T\boldsymbol{\Sigma}_2 \\ -\boldsymbol{\Sigma}_2\boldsymbol{F}(\boldsymbol{I} + \boldsymbol{F}^T\boldsymbol{F})^{-1}\boldsymbol{\Sigma}_1 & \boldsymbol{\Sigma}_2^2 \end{bmatrix}$$

Note that $(-\boldsymbol{\Sigma}_1(\boldsymbol{I} + \boldsymbol{F}^T\boldsymbol{F})^{-1}\boldsymbol{F}^T\boldsymbol{\Sigma}_2)^T = -\boldsymbol{\Sigma}_2\boldsymbol{F}(\boldsymbol{I} + \boldsymbol{F}^T\boldsymbol{F})^{-1}\boldsymbol{\Sigma}_1$, therefore, according to Halko et al. (2011) we have

$$||\boldsymbol{\Sigma}^T(\boldsymbol{I} - \boldsymbol{P_Z})\boldsymbol{\Sigma}|| \leq ||\boldsymbol{\Sigma}_1\boldsymbol{F}^T\boldsymbol{F}\boldsymbol{\Sigma}_1|| + ||\boldsymbol{\Sigma}_2^2|| = ||\boldsymbol{\Sigma}_2||^2 + ||\boldsymbol{F}\boldsymbol{\Sigma}_1||^2 = ||\boldsymbol{\Sigma}_2||^2 + ||\boldsymbol{\Sigma}_2\boldsymbol{\Omega}_2\boldsymbol{\Omega}_1^+||^2$$

Combining the above arguments, we complete the proof. $\qquad\square$

### A.4 PROOF OF THEOREM 3.4

Our method is similar to RSVD in that it employs a sampling matrix $\boldsymbol{\Omega}$ to sample the row space of $\boldsymbol{A}$, approximately capturing the top-$k$ left singular vectors of $\boldsymbol{A}$. Consequently, the result of Theorem 3.3 plays a crucial role, motivating our effort to revise and present a corrected proof of it.

Theorem 3.3 states that when $\boldsymbol{\Omega}_1$ has full column rank, the approximation error admits a deterministic upper bound:

$$||\boldsymbol{A} - \boldsymbol{Q}\boldsymbol{Q}^T\boldsymbol{A}||^2 \leq ||\boldsymbol{\Sigma}_2||^2 + ||\boldsymbol{\Sigma}_2\boldsymbol{\Omega}_2\boldsymbol{\Omega}_1^+||^2$$

By a simple relaxation we further obtain:

$$||\boldsymbol{A} - \boldsymbol{Q}\boldsymbol{Q}^T\boldsymbol{A}|| \leq ||\boldsymbol{\Sigma}_2|| + ||\boldsymbol{\Sigma}_2\boldsymbol{\Omega}_2\boldsymbol{\Omega}_1^+||$$

In the classical RSVD method, the sampling matrix $\boldsymbol{\Omega}$ is taken to be a standard Gaussian matrix, so $\boldsymbol{\Omega}_1$ is almost surely of full row rank, the upper bound for RSVD can be directly derived by invoking the conclusion of Theorem 3.3 and applying suitable relaxations based on the properties of standard Gaussian matrices. However, in our method, $\boldsymbol{\Omega}$ relies on uniform random sampling from $\boldsymbol{A}$, and thus $\boldsymbol{\Omega}_1$ is not guaranteed to be of full row rank under all realizations.

Our proof strategy is to first analyze the quantities $||\boldsymbol{\Sigma}_2\boldsymbol{\Omega}_2\boldsymbol{\Omega}_1^+||$ and $\boldsymbol{\Omega}_1$, and then define a favorable event accordingly. This favorable event occurs with high probability and simultaneously satisfies Theorem 3.3, while its complement occurs with low probability. We employ Theorem 3.3 to bound the expectation of $||\boldsymbol{A} - \boldsymbol{Q}\boldsymbol{Q}^T\boldsymbol{A}||$ when restricted to the favorable event, and use probability arguments to control the expectation restricted to the complement of the favorable event. In this way, we obtain an upper bound on $\mathbb{E}||\boldsymbol{A} - \boldsymbol{Q}\boldsymbol{Q}^T\boldsymbol{A}||$.

First, we analyze the quantities $||\boldsymbol{\Sigma}_2\boldsymbol{\Omega}_2\boldsymbol{\Omega}_1^+||$ and $\boldsymbol{\Omega}_1$. We define the sampling matrix as $\boldsymbol{R}_{l\times m}$, then

$$\boldsymbol{\Omega} = (\boldsymbol{R}\boldsymbol{A})^T = \boldsymbol{A}^T\boldsymbol{R}^T = (\boldsymbol{U}\boldsymbol{\Sigma}\boldsymbol{V})^T\boldsymbol{R}^T = \boldsymbol{V}\boldsymbol{\Sigma}^T\boldsymbol{U}^T\boldsymbol{S}^T$$

Therefore, according to the previous definitions, we have $\boldsymbol{\Omega}_1 = [\boldsymbol{\Sigma}_1, \boldsymbol{0}]\boldsymbol{U}^T\boldsymbol{R}^T$, $\boldsymbol{\Omega}_2 = [\boldsymbol{0}, \boldsymbol{\Sigma}_2, \boldsymbol{0}]\boldsymbol{U}^T\boldsymbol{R}^T$. Thus,

$$||\boldsymbol{\Sigma}_2\boldsymbol{\Omega}_2\boldsymbol{\Omega}_1^+|| = ||\boldsymbol{\Sigma}_2[\boldsymbol{0}, \boldsymbol{\Sigma}_2, \boldsymbol{0}]\boldsymbol{U}^T\boldsymbol{R}^T([\boldsymbol{\Sigma}_1, \boldsymbol{0}]\boldsymbol{U}^T\boldsymbol{R}^T)^+||$$

In order to extract $\boldsymbol{\Sigma}_1^+$ from $([\boldsymbol{\Sigma}_1, \boldsymbol{0}]\boldsymbol{U}^T\boldsymbol{R}^T)^+$, we reformulate $||\boldsymbol{\Sigma}_2\boldsymbol{\Omega}_2\boldsymbol{\Omega}_1^+||$ as follows:

$$||\boldsymbol{\Sigma}_2\boldsymbol{\Omega}_2\boldsymbol{\Omega}_1^+|| = ||\boldsymbol{\Sigma}_2(\boldsymbol{\Sigma}_2\boldsymbol{U}_{n-k}^T\boldsymbol{R}^T)(\boldsymbol{\Sigma}_1\boldsymbol{U}_k^T\boldsymbol{R}^T)^+|| = ||\boldsymbol{\Sigma}_2(\boldsymbol{\Sigma}_2\boldsymbol{U}_{n-k}^T\boldsymbol{R}^T)(\boldsymbol{U}_k^T\boldsymbol{R}^T)^+\boldsymbol{\Sigma}_1^+||$$

where $\boldsymbol{U}_k$ denotes the first $k$ rows of $\boldsymbol{U}$, and $\boldsymbol{U}_{n-k}$ denotes the $(k+1)$-th through $n$-th rows of $\boldsymbol{U}$. By a simple rescaling, we obtain

$$||\boldsymbol{\Sigma}_2\boldsymbol{\Omega}_2\boldsymbol{\Omega}_1^+|| \leq ||\boldsymbol{\Sigma}_2||||(\boldsymbol{\Sigma}_2\boldsymbol{U}_{n-k}^T\boldsymbol{R}^T)(\boldsymbol{U}_k^T\boldsymbol{R}^T)^+||||\boldsymbol{\Sigma}_1^+||$$

$$= \frac{\sigma_{k+1}(A)}{\sigma_k(A)}||(\boldsymbol{\Sigma}_2\boldsymbol{U}_{n-k}^T\boldsymbol{R}^T)(\boldsymbol{U}_k^T\boldsymbol{R}^T)^+||$$

$$\leq ||\boldsymbol{\Sigma}_2(\boldsymbol{U}_{n-k}^T\boldsymbol{R}^T)(\boldsymbol{U}_k^T\boldsymbol{R}^T)^+||$$

We now define the good event. For any $\epsilon \in (0, 1/2)$, we define

$$\mathcal{G}(\epsilon) = \{\sigma_{min}(\boldsymbol{U}_k^T) \geq \sqrt{1-\epsilon} \wedge ||\boldsymbol{U}_{n-k}^T\boldsymbol{R}^T|| \leq 1+\epsilon\}$$

Recall the notion of coherence: for an orthonormal matrix $\boldsymbol{U}$, its coherence is defined as $\mu_k := \frac{m}{k} \max_{1\leq i\leq m} ||(\boldsymbol{U}_k)_{i,:}||_2$.

According to the classical matrix Chernoff–type results (Tropp et al., 2015; Cohen et al., 2020), for every $\delta \in (0, 1)$, if $l \geq C\frac{\mu_k k}{\epsilon^2} \log \frac{k}{\delta}$, then $\mathbb{P}(\mathcal{G}(\epsilon)) \geq 1 - \delta$.

Based on the subspace principal angle lemma and the conclusions of previous work (Magdon-Ismail, 2010; Cohen et al., 2016), we can further deduce that:

$$\mathbb{E}||(\boldsymbol{U}_{n-k}^T\boldsymbol{R}^T)(\boldsymbol{U}_k^T\boldsymbol{R}^T)^+|| \leq C\sqrt{\frac{\mu_k k}{l}}$$

Thus, $\mathbb{E}||\boldsymbol{\Sigma}_2\boldsymbol{\Omega}_2\boldsymbol{\Omega}_1^+|| \leq (C\sqrt{\frac{\mu_k k}{l}})\sigma_{k+1}(\boldsymbol{A})$. $\mathbb{E}||\boldsymbol{A} - \boldsymbol{Q}\boldsymbol{Q}^T\boldsymbol{A}|| \leq (1 + C\sqrt{\frac{\mu_k k}{l}})\sigma_{k+1}(\boldsymbol{A})$, which is the expectation of $||\boldsymbol{A} - \boldsymbol{Q}\boldsymbol{Q}^T\boldsymbol{A}||$ when restricted to the favorable event.

Finally, we derive the expectation restricted to the complement of the favorable event. It is evident:

$$\mathbb{E}||\boldsymbol{A} - \boldsymbol{Q}\boldsymbol{Q}^T\boldsymbol{A}|| = \mathbb{E}||(\boldsymbol{I} - \boldsymbol{P}_{\boldsymbol{Y}})\boldsymbol{A}|| \leq \mathbb{E}||\boldsymbol{A}|| = ||\boldsymbol{A}||\mathbb{P}(\mathcal{G}^c) \leq \delta||\boldsymbol{A}|| \leq ke^{\frac{-l\epsilon^2}{C\mu_k k}}||\boldsymbol{A}||.$$

Since $\epsilon$ is an arbitrary positive real number, we may set $\epsilon = \sqrt{\frac{\mu_k k}{l}}$. Substituting this choice yields:

$$\delta||\boldsymbol{A}|| \leq ke^{\frac{1}{C}}||\boldsymbol{A}||$$

By adding the expectation of $||\boldsymbol{A} - \boldsymbol{Q}\boldsymbol{Q}^T\boldsymbol{A}||$ restricted to the favorable event and the expectation restricted to its complement, we obtain the desired conclusion. $\qquad\square$

## A.5 RELATED WORK

**Parameter-Efficient Fine-Tuning** Parameter-efficient fine-tuning (PEFT) provides an effective strategy for adapting pre-trained models to downstream tasks. By modifying only a small subset of parameters, these methods achieve competitive performance compared with full fine-tuning. The additive PEFT methods, like Adapter Tuning (Houlsby et al., 2019), Prompt Tuning (Lester et al., 2021) and Prefix Tuning (Li & Liang, 2021), keep the original model weights frozen and introduce new trainable components, but these approaches increase inference latency. Reparameterization-based PEFT methods, such as Low-Rank Adaptation (LoRA) (Hu et al., 2022), operates by constraining trainable parameters within a low-rank subspace for each weight matrix. Besides, SPDF (Thangarasa et al., 2023), S$^2$FT (Yang et al., 2024), and SMT (He et al., 2025) leverage sparsity by selectively training only a subset of the model weights. However, a significant memory bottleneck still remains for existing PEFT methods, as these methods do not reduce the memory required for storing activations, thereby limiting their practicability in resource-constrained scenarios.

**Parameter Quantization and Decomposition**  The growing demand for computational efficiency highlights the deployment challenges posed by the large size of modern foundation models, particularly on edge devices. Model quantization and decomposition serve as the principal strategies to mitigate this challenge. Existing quantization techniques generally fall into two categories: outlier-aware weight-only quantization (Lin et al., 2024; Lee et al., 2024) and joint activation-weight quantization (Li et al., 2024; Hu et al., 2025). Regarding model decomposition, a straightforward application of SVD often harms performance, since weight matrices tend to be high-rank (Yu & Wu, 2023). To overcome this, FWSVD (Hsu et al., 2022), ASVD (Yuan et al., 2023), and WeLore (Jaiswal et al., 2025) decompose weights into low-rank matrices by preserving output similarity based on activations. LoSparse (Li et al., 2023) aim to recover high-rank information by additionally augmenting a low-rank component with a sparse matrix. The further approaches combine decomposition with quantization, representing weights as the sum of a low-rank component and a quantized matrix (Guo et al., 2023). Besides these inference-time optimizations, QLoRA (Dettmers et al., 2023) and LQ-LoRA (Guo et al., 2023) reduce memory consumption for fine-tuning by first quantizing the pre-trained weights and then applying low-rank adaptation. However, the quantization and decomposition methods inevitably lead to a loss in model performance.

**Gradient and Optimizer State Compression**  To reduce the memory consumption during training processes, recent work focuses on compressing gradients and optimizer states, considering they are naturally low-rank during training, which has been studied in both theory and practice (Cosson et al., 2023; Shen et al., 2025). Among existing approaches, GaLore (Zhao et al., 2024) and FLoRA (Hao et al., 2024) project gradients into a low-rank subspace for optimizer updates, thereby saving the memory cost. However, GaLore relies on SVD for constructing the projection matrix, which introduces non=negligible computational overhead. Moreover, the low-rank updating subspace is discontinuous. Subsequent research aims to mitigate these issues by improving the projection matrix (Muhamed et al., 2024; Zhang et al., 2025; Zhao et al., 2025; Xiao et al., 2025; Liang et al., 2024) or integrating GaLore with other optimizer-focused techniques (Huang et al., 2024; Zhu et al., 2024). Different from the projection-based approaches, LoGE (Zhang et al., 2024) uses a low-rank decomposition of weights specifically during the backward pass to compute activation gradients, thus accelerating the fine-tuning efficiency. Nevertheless, the memory consumed by activations is often neglected. This is especially pronounced under the PEFT setting, where activations become the primary memory bottleneck. Therefore, existing strategies focusing solely on gradients and optimizer states are unsuitable for PEFT.

### A.6  LIMITATIONS AND DISCUSSIONS

While LORACT demonstrates significant memory reduction and competitive performance across a wide range of tasks, its effectiveness is sensitive to the chosen compression ratio $r$. In particular, under aggressive compression settings (e.g., $r < 1/8$ for vision tasks), a noticeable decline in accuracy is observed. This suggests that the activation compression may discard some critical information under certain fine-tuning scenarios. This limitation may pose challenges for LORACT in some complex reasoning benchmarks, where the estimation errors may accumulate and then degrade model performance. Therefore, it is crucial to explore adaptive compression strategies with optimal $r$ based on critical information, such as activation characteristics or task-specific requirements, and we leave this problem for future research.

