# OpenReview forum: "Memory-Efficient Fine-Tuning via Low-Rank Activation Compression"
_ICLR.cc/2026/Conference — Submitted to ICLR 2026_

### Official Review · Reviewer_d1h8 · 2025-10-27

**Soundness:** 2
**Presentation:** 3
**Contribution:** 2
**Rating:** 2
**Confidence:** 5

**Summary:**

The paper establishes a compression of parameter efficient fine tunning methods by compressing the model activations. They show that model activations constitute to a large memory source in parameter efficient fine tuning methods that is often overlooked when such methods are applied to fine tuning foundation models. Motivated by this the authors introduce Low-Rank Activation Compression (LoRACT) that provides a compression strategy for model activations leading to further memory compression in parameter efficient fine tunning methods. The authors prove theorems that shows that LoRACT can still maintain bounded error when compared to the case of no compression on the model activations. They then apply LoRACT in the context of vision fine tuning and language fine tuning. In both cases the authors show that their methodology does performs better than baseline parameter efficient fine tuning methods yet can lead to around 80% less activation memory.

**Strengths:**

**Originality:** The main concept of the paper on compressing model activations in parameter efficient fine tunning methods is nice. However, activation compression has been done on models where full pre-training is being done such as in Compressing the Activation Maps in Deep Neural Networks and Its Regularizing Effect, TMLR 2024, by Vu et al. Thus I am not surprised one can compress activations in parameter efficient fine tunning methods as well. The method the authors introduce for this compression technique is via an online matrix compression algorithm that is carried out at every forward pass which I have never seen done before.

**Clarity:** In general the paper is well written and the main ideas are explained well. I found it easy to read and had no issues with how their methodology was presented. One thing I would have liked to see is more discussion about related work in the main paper. The authors have a discussion of related work in the appendix. For example, activation compression has been carried out when pre-training deep learning models, for example as in  Compressing the Activation Maps in Deep Neural Networks and Its Regularizing Effect, TMLR 2024, by Vu et al, and it would have been good if the authors positioned their work in the context of such works. This would help readers understand how their work is positioned in the broader area of pre-training and fine tuning deep networks. While they do have some discussion in paragraph 3 of the introduction I would think a small related work section in the main paper would have led to more clarity.

**Weaknesses:**

**Novelty:** My main issue with the paper is the novelty. The idea of compressing activations is not new in deep learning models and has been done before in papers such as Compressing the Activation Maps in Deep Neural Networks and Its Regularizing Effect, TMLR 2024, by Vu et al, a google search will provide many other papers. Furthermore, in the context of fine tuning there have been works that have compressed activations though in a slightly different way. For example, CompAct: Compressed Activations for Memory-Efficient LLM Training by Shamshoum et al, uses low-rank compressed activations for the backward pass and thus also reduces memory in the context of LLMs. The work HyC-LoRA: memory efficient LoRA Fine-Tuning with hybrid activation compression by Wang et al also applies a type of activation compression within the context of fine tuning. Given that there are such works on activation compression I would say the idea of the authors to apply activation compression to PEFT methods is nothing surprising nor novel.

**Significance:** I unfortunately don't think the results of the paper are of great significance. I also don't think the paper will have a great impact on the community. My main reason for thinking this is that the method is very simply and is more on the side of engineering than mathematics. In fact, although the authors provide some theorems that clearly show their compression leads to a loss that remains a bounded difference from the uncompressed version this theorem (namely theorem 3.1) is an extremely simple consequence of Lipshitz continuity and I don't believe it warrants a theorem nor is it significant.

**Questions:**

1. The statement of theorem 3.1 is supposed to give the reader the idea that the loss on $F_{comp}$ shouldn't be too far from the loss of $F$. However, this is only true if the right hand side of the inequality in theorem 3.1 is small. This in turn depends on the Lipshitz constants. Going to the proof on p.14 what happens if $\sigma_1$ is large? This will then give a really bad bound on the right hand side of theorem 3.1. Do we know apriori that $\sigma_1$ would be small? The bound in the right hand side of the theorem 3.1 is additive and so the same argument applies for each summand. How do you know each summand is small? Furthermore, if $N$ is large then these summands, although small, can add up to something large. You should say something about this.

2. In Theorem 3.4 the bound you get depends on the k-coherence of the matrix $A$ which you rightly point out and then you say therefore the bound is tight. However, the bound also depends on the singular value $\sigma_{k+1}$. So I am guessing for the bound to be tight you would also need $\sigma_{k+1}$ to be small. Can you explain why this is the case?

3. The proof of Theorem 3.4 in the appendix was difficult for me to follow. On line 889 you write "based on the subspace principal angle lemma and the conclusion of previous works...". What conclusions from the previous works that you cite are you using? Please can you state this clearly for me. Also, it would be very helpful for the reader if you stated the subspace principal angle lemma before the proof of the theorem and then referred to it so that the reader can refer to it if they need to.

4. I noticed in the experiments section you ran your experiments against the original LoRA from Hu et al. However, there have been several variants since then that do a much better process of PEFT on foundation models such DoRA by Liu et al., RandLoRA by Albert et al. Why didn't you try your method on more recent PEFT methods?

5. In the vision transformer experiments you run your experiments on ViT-B from Dosovitskiy et al. Since then there have been several more vision transformers that have shown to achieve a much better performance on a variety of image classification tasks such as XCiT by El-Nouby et al., Swin transformer by Liu et al. Does your method just naturally extend to these more modern vision transformers?

---

### Official Review · Reviewer_pZrM · 2025-10-28

**Soundness:** 2
**Presentation:** 3
**Contribution:** 2
**Rating:** 4
**Confidence:** 4

**Summary:**

The authors note that in PEFT settings such as LoRA, activations remain the dominant memory bottleneck and exhibit a stable low-rank structure. They propose LORACT for online activation compression: after the forward pass, the activations needed for backpropagation are factorized, only the compact factors are stored, and activations are reconstructed during the backward pass to reduce activation memory. To improve efficiency, they replace randomized SVD with a sampling-based orthogonal factorization, provide a theoretical error bound, and introduce a pre-norm Transformer adaptation in which the Norm output and the sublayers share the same compressed activations. Experiments on language tasks (Alpaca, FLAN-v2, WikiText-2) and vision tasks (CIFAR-100, Food-101) indicate that, over a reasonable range, LORACT can save about 80% additional activation memory relative to LoRA, with comparable performance on language tasks and only slight degradation on vision tasks under conservative compression ratios.

**Strengths:**

（1）Demonstrates that, under LoRA, activation memory—not gradients or optimizer states—dominates, and scales linearly with batch size and context length, stressing consumer-grade GPUs.
（2）Uses the Norm output as a single shared activation to store/compress, with a clear mechanism to backpropagate from it, reducing both memory footprint and unnecessary computation.

**Weaknesses:**

（1）The paper cites LoRA-FA, CompAct, ESPACE, SoLA, and CoLA, but does not compare against them experimentally. Only comparing to vanilla LoRA makes it hard to position the method’s relative advantages in accuracy and speed. At minimum, reproducible comparisons to SoLA/CompAct/CoLA are needed.
（2）Key baselines are absent:SoLA / CoLA / CompAct (compressed or sparse activations, or activation-side alternatives to LoRA). Activation checkpointing (pure checkpointing vs. the proposed online compression, and their combination). RSVD / random projections as drop-in replacements for the proposed compression module (compute vs. error vs. stability).
（3）The paper should report per-step latency, throughput, total training time, and peak memory for all methods, under a unified implementation setting (e.g., same AMP and checkpointing configurations).
（4）Spelling errors such as “establishe” should be corrected.

**Questions:**

The paper compares only against vanilla LoRA and lacks horizontal evaluations against improved LoRA variants (AdaLoRA, DyLoRA, LoRA-FA, DoRA/VeRA/Compacter, QLoRA) and activation-side approaches (SoLA/CoLA/CompAct, activation checkpointing), making it hard to identify the method’s relative advantages. Task coverage is also limited: please add mathematical reasoning (MATH, GSM8K), general reasoning (BBH, ARC-C), and Stable Diffusion image generation (reporting FID/CLIP score), and present quality–memory–latency Pareto curves under long-context settings. All baselines should be compared under two fairness criteria—equal training steps and equal peak memory—while also reporting per-step latency, throughput, and total training time. Provide systematic ablations on rank selection strategies, the compression operator (sampling-based factorization vs. RSVD/random projections), and combinations with QLoRA. If these critical comparisons are added, the work would have strong practical value for single-GPU efficient fine-tuning; in its current form, the evidential support remains insufficient.

---

### Official Review · Reviewer_RFBG · 2025-10-29

**Soundness:** 2
**Presentation:** 2
**Contribution:** 2
**Rating:** 4
**Confidence:** 3

**Summary:**

To mitigate the memory usage in fine-tuning process, this paper proposes a memory-efficient fine-tuning approach Low-Rank Activation Compression (LoRA-CT). It provides a more flexible and versatile compressing strategy that can be applied online during the forward
pass without the need for any calibration data. Moreover, LoRA-CT incorporates a novel sampling-based orthogonal decomposition algorithm specifically designed for low-rank matrices. Experiments are carried out to demonstrate the superiority of the propose methods.

**Strengths:**

1. This paper provide a valuble insight in the LLM fine-tuning process that the activation has low-rank structure and can be compressed to save memory.

2. Efficiency gains: LoRA-CT is shown to reduce memory consumption, indicating practical benefits for resource-constrained scenarios.

3. Ablation study and analysis is good and convincing. Paper presents in-depth analysis regarding the proposed method and overall assessment is good, providing a convincing evidence to demonstrate the superiority of the proposed method. However, there are few important concerns needed to be addrees, see Weaknesses belows.

**Weaknesses:**

1. Evaluation is limited. The experiments should involves other baselines such as QLoRA, and other state-of-the-art PEFT methods, and latest LLMs models (llama-3-8B) instead of their predecessors and other outdated models. Besides, the downstream tasks are limited, authors should explore the performance of the proposed method in other downtream tasks.

2. Experiment description is insufficient. To ensure a fair evaluation and comparison, authors should report all training setups that can have implication in the final model performance, such how many epoch in training, is there any learning scheduler, and etc. Additionally, the comparison in this paper should also include training time to faithfully report the training process.

3. Paper is hard to follow. Some theorems (3.1 and 3.2) lack reference in the text, which makes them orphans in this paper and seems there no relavant text related to the theorems.

**Questions:**

See Weaknesses above.

---

### Official Review · Reviewer_MRKR · 2025-10-31

**Soundness:** 3
**Presentation:** 3
**Contribution:** 3
**Rating:** 6
**Confidence:** 4

**Summary:**

This paper addresses the problem of high activation memory consumption in Parameter-Efficient Fine-Tuning (PEFT) methods. The authors propose Low-Rank Activation Compression (LoRACT), a method that leverages the empirically observed low-rank nature of activations. LoRACT performs an online, calibration-free compression of activation matrices during the forward pass, storing only their low-rank factors, which are then used to reconstruct the activations for the backward pass. The paper introduces a new sampling-based decomposition algorithm for efficiency and a specialized "pre-norm" compression strategy for Transformer models. Extensive experiments on language and vision tasks show that LoRACT can reduce activation memory by approximately 80% compared to LoRA while maintaining competitive, and sometimes even superior, performance.

**Strengths:**

The paper correctly identifies activation memory as the dominant cost in PEFT (nicely illustrated in Figure 1) and provides convincing empirical evidence for its core assumption—the low-rank structure of activations (Figure 2).

The evaluation is thorough, covering both language and vision domains, and using standard benchmarks and models. The inclusion of results with varying compression ratios (r) provides a clear picture of the trade-offs.

The comparison in Tables 3, 4, and 6 between the proposed "pre-norm" strategy and a more naive "layer-wise" approach clearly demonstrates the superiority of the tailored design, strengthening the paper's claims.

**Weaknesses:**

The paper only compares LoRACT against LoRA. No direct comparison is provided against LoRA-FA, CompAct , CoLA or VeLoRA (cited lines 53-65). This comparison would have help substantiate the trade-offs of LoRACT's decomposition versus these other approaches.

The paper quickly dismisses gradient compression methods like GaLore as "ill-suited for the PEFT setting." A more detailed discussion or even a combined experiment (e.g., LoRA + GaLore vs. LoRACT) would provide a more holistic view of where LoRACT fits into the broader landscape of memory-efficient training techniques.

The paper lacks any wall-clock time or runtime analysis. The paper claims its novel decomposition algorithm is more computationally efficient than RSVD, but this is only supported by theoretical arguments (i.e., avoiding the generation of a Gaussian matrix). It is crucial to understand the actual computational overhead (latency) introduced by the online decomposition during training. Without this, the full trade-off between memory savings and training speed remains unclear.

The performance of LoRACT is sensitive to the compression ratio r. The paper does not provide a clear methodology or intuition for selecting r for a new task or model, other than treating it as a hyperparameter to be tuned. A deeper analysis of effective rank trends for activations across different models and layers would make the method more practical to apply.

**Questions:**

Could the authors provide an simple empirical analysis of the training throughput (e.g., wall-clock time per epoch or samples per second) for LoRA versus LoRACT at different compression ratios?

In some cases (e.g., Alpaca with r=1/16), LoRACT slightly outperforms LoRA. The paper speculates this may "facilitate the activation quality." Could you elaborate on this hypothesis? Is it possible that the low-rank projection acts as a form of regularization by filtering out noise in the activations?

How was the compression ratio r determined for the experiments? Was a single value of r used for all compressed layers within the model? Have the authors explored layer-specific compression ratios, and do the authors have any insights into how the optimal r might vary across layers or model architectures?

In the proposed, pre-norm strategy, the algorithm stores the RMS value for the backward pass. Could the authors quantify its memory footprint relative to the stored low-rank matrices U and V to empirically confirm that its cost is negligible as stated?

---

### Meta-Review · Area_Chair_zeNB · 2026-01-06

**Summary:**

Reviewers agree that the paper targets an important and timely bottleneck(activation memory in PEFT) and that the proposed online low-rank activation compression idea is clearly motivated and empirically promising in the reported settings (large memory savings with mostly competitive accuracy, plus a Transformer-tailored pre-norm variant and some theoretical bounds). However, multiple reviewers conclude that the current evidence is insufficient for acceptance because the evaluation is not well-positioned against the most relevant alternatives: experiments are largely limited to vanilla LoRA, omitting key PEFT variants and activation-side baselines that are directly related and even cited (e.g., LoRA-FA/SoLA/CoLA/CompAct/VeLoRA/QLoRA), making it difficult to assess true relative advantages. A second recurring concern is the absence of wall-clock/runtime analysis (throughput/latency/total training time) to quantify the compute–memory trade-off introduced by online decomposition, which is central to the method's practical value. Reviewers also request clearer guidance on choosing the compression ratio/rank and whether it should be layer-specific, broader task/model coverage to support generality, and stronger novelty/positioning given prior activation compression work. Finally, some concerns remain about reproducibility and presentation (insufficient training setup details, limited in-text integration of theorems/clarity of proofs, and minor writing issues). Since no rebuttal was provided, none of these issues were clarified or resolved, which supports a reject decision despite the promising direction.

**Reviewer Concerns:**

Addressed by rebuttal:
* None (no rebuttal was submitted).

**Reviewer Scores:**

N/A The authors have not participated in the rebuttal process.

---

### Decision · Program_Chairs · 2026-01-26

Reject